# An Efficient Quantum Classifier Based on Hamiltonian Representations

## Abstract

Quantum computing shows great potential for expanding the range of efficiently solvable problems. This promise arises from the advantageous resource and runtime scaling of certain quantum algorithms over classical ones. Quantum machine learning (QML) seeks to extend these advantages to data-driven methods. Initial evidence suggests quantum-based models can outperform classical ones in terms of scaling, runtime and generalization capabilities. However, critics have pointed out that many works rely on extensive feature reduction or use toy datasets to draw conclusions, raising concerns about their applicability to larger problems. Scaling up these results is challenging due to hardware limitations and the high costs generally associated with encoding dense vector representations on quantum devices. To address these challenges, we propose an efficient approach called *Hamiltonian classifier* inspired by ground-state energy optimization in quantum chemistry. This method circumvents the costs associated with data encoding by mapping inputs to a finite set of Pauli strings and computing predictions as their expectation values. In addition, we introduce two variants with different scaling in terms of parameters and sample complexity. We evaluate our approach on text and image classification tasks, comparing it to well-established classical and quantum models. Our results show the Hamiltonian classifier delivers performance comparable to or better than these methods. Notably, our method achieves logarithmic complexity in both qubits *and* quantum gates, making it well-suited for large-scale, real-world applications.

## 1 Introduction

In recent years, interest in quantum computing has grown significantly due to the provable advantages in computational complexity and memory usage some algorithms exhibit over their best classical analogues (Nielsen & Chuang, 2000; Quetschlich et al., 2024; Biamonte et al., 2017). For instance, efficient algorithms exist that can solve problems such as integer factorization (Shor, 1997), Fourier transform (Camps et al., 2021), and specific instances of matrix inversion (Harrow et al., 2009) with an exponential speedup over the fastest known classical methods. Other notable results include Grover's algorithm, which performs an unstructured search achieving a $O(\sqrt{n})$ time complexity, a quadratic improvement over the fastest classical approach that scales as $O(n)$ (Grover, 1996). In parallel with these theoretical developments, the size of publicly accessible quantum machines has been steadily growing, and several companies have begun offering commercial cloud access to these devices (Yang et al., 2023). Quantum machine learning is an offshoot of quantum computing that seeks to extend its advantages to data-driven methods. The leading paradigm revolves around variational quantum circuits (VQCs), quantum algorithms whose parameters can be adjusted with classical optimization to solve a specific problem (Cerezo et al., 2021). This approach is sometimes referred to as quantum neural networks (QNNs) given the similarities with the classical counterpart (Farhi & Neven, 2018; Killoran et al., 2019). Prior works have found some evidence that QML algorithms can offer improvements over their classical analogues in terms of capacity (Abbas et al., 2021), expressive power (Du et al., 2020), and generalization capabilities (Caro et al., 2022).

Despite the advancements of VQCs, a large-scale demonstration of *advantage* - the ability of a quantum computer to solve a problem faster, with fewer resources or better performance than any classical counterpart - remains out of reach for QML algorithms. Current quantum machines, referred

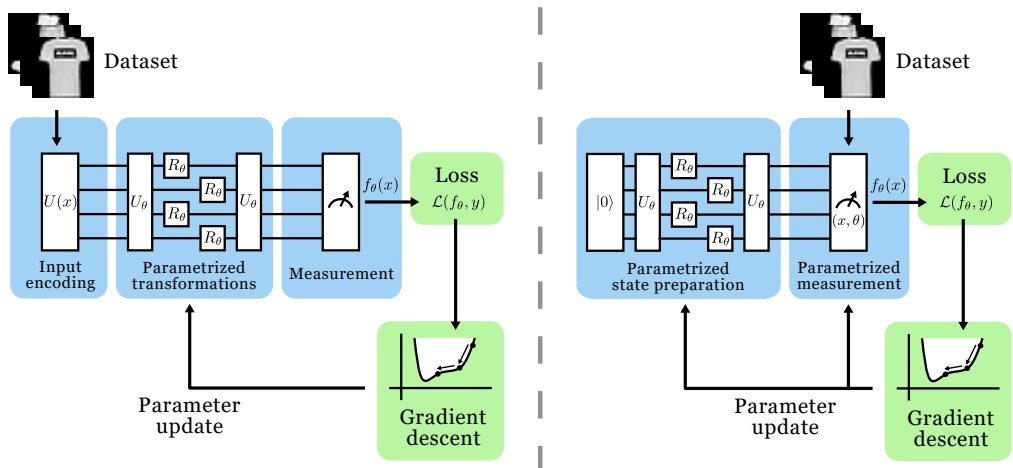

Figure 1: Comparison between the standard VQC training loop (left) and ours (right). In green, components evaluated classically; in blue, components evaluated on quantum computers.

to as *Noisy Intermediate-Scale Quantum* (NISQ) devices (Preskill, 2018), are restricted both in size and complexity of the operations they can perform (Zaman et al., 2023). *Qubits*, the basic units of quantum computation, are hard to maintain in a state useful for computation, and scaling quantum processors to a size suitable for QML remains a significant challenge. Moreover, the widespread use of dense and unstructured vector representations in modern machine learning architectures (Bengio et al., 2000) makes their translation into quantum equivalents challenging. Specifically, the processes of loading data onto a quantum device, known as *encoding*, and extracting results from it, referred to as *measurement*, can scale rapidly in terms of qubit requirements or computational costs (Schuld & Killoran, 2022). These costs may grow prohibitively with input size, potentially negating any quantum advantage. As a result, researchers are often forced to down-scale their experiments making it unclear whether these findings generalize at larger scales (Bowles et al., 2024; Mingard et al., 2024).

Motivated by these limitations, we propose a scheme to efficiently encode and measure classical data from quantum devices and demonstrate its effectiveness across different tasks. The method we propose is a specific instance of a *flipped model* (Jerbi et al., 2024), a type of circuit which encodes data as the observable of a quantum circuit rather than as a quantum state, effectively bypassing the need for input encoding. From a linear algebra perspective, this corresponds to learning a vector representation of the classification problem and using the input data to compute projections of this vector. The magnitude of these projected vectors is then used to make predictions. In quantum-mechanical terms, unstructured input data is mapped onto Pauli strings which are then combined to construct a Hamiltonian. The classifier prediction is obtained as the expectation value of this Hamiltonian. This idea shares many similarities with variational quantum eigensolver (VQE), a type of VQC widely used in quantum chemistry for solving the electronic structure problem (Peruzzo et al., 2014; Tilly et al., 2022). We improve over the standard flipped model by providing a mapping from inputs to observables that achieves a favourable logarithmic qubit *and* gate complexity relative to the input dimensionality. We also introduce two variants that trade some classification performance in exchange for smaller model size and better sample complexity, respectively. Most importantly, the constant scaling in sample complexity of the second variant allows it to be efficiently adapted for execution on quantum devices. In brief, this paper provides the following three contributions:

$C_1$ A novel encoding scheme achieving logarithmic qubit and gate complexity;

$C_2$ A thorough evaluation of our scheme on text and image classification tasks;

$C_3$ An empirical and theoretical comparison of our method against other established quantum and classical baselines.

Section 2 provides an overview of data encoding, discusses the current state of QML, and describes the key issues preventing large-scale experimentation from being performed. To address these challenges, we introduce in Section 3 our novel quantum encoding architecture. Specifically, in Section 3.6, we provide a theoretical comparison of the scaling behaviour of our method relative to other quantum-based approaches. In Section 4 we benchmark its effectiveness against other quantum and classical baselines, obtaining promising results. Our chosen datasets are SST, AG News, IMDb, MNIST, Fashion-MNIST, and CIFAR, covering text and image domains in both the binary and multi-class scenarios. Finally, in Section 5, we summarize our findings and discuss future directions.

## 2 PRELIMINARIES

Quantum computers differ fundamentally from classical computers, utilizing the principles of quantum physics rather than binary logic implemented by transistors. For readers unfamiliar with the topic, in Appendix A we offer a brief introduction to the notation and formalize the concepts of qubit, gate and measurement. In this section, we instead discuss how these devices have been used to tackle machine learning problems, as well as their limitations.

### 2.1 DATA ENCODING

The first step for quantum-based ML algorithms is representing input data as quantum states, a process also known as *state preparation*. Most algorithms share similarities in how they achieve this (Rath & Date, 2023). The choice of encoding severely impacts the runtime of the circuit as well as its expressivity (Sim et al., 2019). *Basis encoding* is the simplest representation analogous to classical bits. An $n$-element bit string $[x_1 \; x_2 \; \ldots \; x_n]$ is represented in the basis states of $n$ qubits as $\bigotimes_{i=1}^{n} |x_i\rangle = |x_1 \ldots x_n\rangle$ (e.g. 011 is represented as $|011\rangle$). This representation requires $O(n)$ qubits. *Angle encoding* represents continuous data as the phase of qubits. A set of $n$ continuous variables $[x_1 \; x_2 \; \ldots \; x_n]$ can be represented over $n$ qubits as $\bigotimes_{j=1}^{n} R_{\hat{\sigma}}(x_j) |0\rangle$ with $R_{\hat{\sigma}}(x) = e^{-ix\hat{\sigma}}$ a so-called rotation gate, and $\hat{\sigma}$ a Pauli matrix $X, Y$ or $Z$ that specifies the rotation axis. *Amplitude encoding* represents a vector of $N = 2^n$ values $[x_1 \; x_2 \; \ldots \; x_N]$ over $n$ qubits as $\sum_{i=0}^{N-1} x_i |i\rangle$, where $\{|i\rangle\}_{i=1}^{N}$ corresponds to the canonical orthonormal basis written in a binary representation.

There is a trade-off between ease of encoding and qubit count: basis and angle encoding use $O(n)$ gates for $O(n)$ values over $O(n)$ qubits, while amplitude encoding handles $N$-dimensional inputs but needs $O(N)$ gates over $O(n)$ qubits, an exponential increase in input size but also gates. Angle encoding strategies often embed multiple features onto the same qubit to reduce qubit usage, effectively a form of pooling (Pérez-Salinas et al., 2020; Du et al., 2020). Amplitude encoding, on the other hand, is often performed using easy-to-prepare quantum states to mitigate its gate complexity (Ashhab, 2022; Du et al., 2020). In text-related tasks, encoding schemes typically encode words over a set number of qubits (Wu et al., 2021; Lorenz et al., 2021), while in image tasks, pixel values are directly encoded as angles or amplitudes (Cong et al., 2019; Wei et al., 2022).

### 2.2 VARIATIONAL QUANTUM CIRCUITS

Once data has been encoded in a quantum computer, it is processed by a quantum circuit to obtain a prediction. One of the leading paradigms for QML revolves around VQCs (also called quantum *ansätze*, or parametrized quantum circuits), a type of circuit where gates are specified by classical parameters. The training loop of a VQC (Figure 1) closely resembles that of classical neural networks (Cerezo et al., 2021): input data is encoded in the quantum device as a quantum state, several layers of parametrized gates transform this state, a prediction is obtained via quantum measurement, and finally a classical optimizer computes a loss and updates the parameters. Specialized optimizers allow backpropagating through a quantum circuit, but parameters are saved classically and the process is otherwise the same (Wierichs et al., 2022). In VQCs applied to machine learning, measurement usually plays the role of feature extractor, producing either features to be further processed by downstream layers (Chen et al., 2022), or the final prediction (Farhi & Neven, 2018). VQCs have shown better convergence properties during training (Abbas et al., 2021) as well as better expressive power (Du et al., 2020) when compared with neural networks of similar size.

Several VQC-based equivalents of classical architectures have been proposed ranging from simple neurons (Cao et al., 2017) to more elaborate schemes like auto-encoders (Romero et al., 2017), generative adversarial networks (Dallaire-Demers & Killoran, 2018), RNNs (Bausch, 2020; Li et al., 2023), attention layers (Cherrat et al., 2022; Shi et al., 2023; Zhao et al., 2024), and convolutional neural networks (Cong et al., 2019; Henderson et al., 2020).

### 2.3 QUANTUM MACHINE LEARNING LIMITATIONS

Despite these achievements, QML applications have yet to demonstrate a practical quantum advantage. Firstly, NISQ devices are limited in terms of the number of qubits in a single device, connectivity between the qubits, noise in the computation, and coherence time (Zaman et al., 2023; Anschuetz & Kiani, 2022). Secondly, quantum devices have fundamental difficulties in dealing with the dense and unstructured vector representations around which ML revolves: loading (or encoding) data into a quantum state either requires a large number of qubits (for angle and basis encoding) or a prohibitive amount of gates (for amplitude encoding). Efficient methods for amplitude encoding (Ashhab, 2022; Wang et al., 2009) incur trade-offs in the expressivity of vectors that have not been explored sufficiently in the context of QML. Moreover, measurement is an expensive process extracting only one bit of information per qubit measured. Extracting a real-valued vector from a quantum computer generally requires exponentially many measurements (Schuld & Petruccione, 2021). As a result, many current QML experiments are limited in both scale and scope in order to fit within the constraints of NISQ devices and simulators. Small datasets, typically consisting of only a few hundred samples, are often used (Senokosov et al., 2024; Li et al., 2023; Liu et al., 2021; Chen, 2022). Additionally, aggressive dimensionality reduction is commonly performed to reduce data to just a few dozen features (Bausch, 2020; Zhao et al., 2024). In contrast, even "small" classical neural networks by modern standards are several orders of magnitude larger in terms of parameters, dataset size, and representation dimensionality (Mingard et al., 2024). These limitations have raised concerns about the applicability of QML results to larger, more complex tasks. Some question whether the observed performance is due to the quantum model itself or the upstream pre-processing (Chen et al., 2021), while others highlight the difficulty of generalizing these results to larger datasets and the challenges of fair benchmarking (Bowles et al., 2024; Mingard et al., 2024).

Several other challenges remain, including how to mitigate barren plateaus (Larocca et al., 2024), develop efficient optimization algorithms for VQCs (Wiedmann et al., 2023), and implement effective error correction schemes(Chatterjee et al., 2023). In this work, we show how limitations pertaining encoding and measurement can be mitigated by changing the way inputs are represented in a quantum device.

## 3 HAMILTONIAN CLASSIFIER

Recognizing the limitations of current encoding techniques and the need for more efficient methods on current quantum devices, we introduce a novel approach: the *Hamiltonian classifier*. Instead of relying on expensive state preparation procedures, our method maps inputs to a set of Pauli strings that measured together yield a binary prediction. This approach is inspired from the Variational Quantum Eigensolver formulation applied to quantum chemistry (Peruzzo et al., 2014) with the key difference that the Hamiltonian is constructed from data instead of being derived from the quantum-physical properties of the chemical system at hand. As in the general VQE setting, we then optimize the objective function with classical methods. Independent works (Jerbi et al., 2024) have introduced the theoretical framework for flipped models, a category that includes our Hamiltonian classifier. It has been proven that certain types of flipped models, particularly when combined with classical shadow techniques, can demonstrate a quantum advantage for specific tasks. Additionally, flipped models have been previously explored in the context of quantum federated learning (Song et al., 2023). Our work distinguishes itself as the first to apply flipped models to text classification and to provide a thorough evaluation of such methods across multiple datasets, comparing against both classical and quantum baselines. Moreover, we enhance the practicality of these methods by introducing an encoding scheme that sensibly lowers qubit requirements. We also propose variants that further reduce model size and sample complexity with minimal impact on performance and extend this method to the multi-class scenario. These factors make the Hamiltonian classifier applicable not only to toy problems but also to real-world tasks.

## 3.1 VARIATIONAL QUANTUM EIGENSOLVERS (VQES)

VQEs are a class of algorithms related to QML which have been extensively utilized in quantum chemistry and condensed matter physics for finding the lowest-energy configuration in quantum systems (Peruzzo et al., 2014; Tilly et al., 2022). VQEs are considered one of the most promising approaches for achieving a practical advantage in the NISQ era (Daley et al., 2022). In practice, VQEs solve the ground state energy problem by approximating the lowest eigenstate of a Hamiltonian $H$, a Hermitian matrix describing a quantum system. This is achieved by classically optimizing a VQC preparing a state $\psi_\theta$ for which the system energy $E := \langle H \rangle$ is minimal: $E_{\min} := \min_\theta \langle \psi_\theta | H | \psi_\theta \rangle$. For the electronic structure problem, these Hamiltonians are constructed from fundamental principles, resulting in a polynomial number of Pauli strings. Note that the vector input $\psi_\theta$ is never expressed explicitly; instead, it is implemented on the quantum computer, serving as a central component in the optimization process.

## 3.2 FULLY-PARAMETRIZED HAMILTONIAN (HAM)

Our Hamiltonian classifier (Fig. 2) takes as input a sequence of embeddings $x = [x_1 \ x_2 \ \ldots \ x_s]$, $x_i \in \mathbb{R}^d$ and outputs a prediction probability $f_{\theta,\phi}$, a real number representing the estimated class the input belongs to. Similarly to VQCs, we use gradient descent to optimize the classifier parameters $\theta, \phi$ on a given training set $\mathbf{X}$ with binary labels $\mathbf{y}$. The classifier can encode embeddings of size at most $N = 2^n$ with the remaining $N - d$ dimensions padded to 0. The optimization problem can be summarized as follows:

$$\arg \min_{\theta,\phi} \frac{1}{|\mathbf{X}|} \sum_{x \in \mathbf{X}, y \in \mathbf{y}} \mathcal{L}(f_{\theta,\phi}(x), y), \tag{1}$$

$$f_{\theta,\phi}(x) := \sigma(\psi_\theta^\dagger H_\phi(x) \psi_\theta) \tag{2}$$

$$H_\phi(x) := H_\phi^0 + \frac{1}{s} \sum_{i=1}^s x_i x_i^\top \tag{3}$$

$$\psi_\theta := U_\theta |0\rangle^{\otimes n} = U_\theta [1 \ 0 \ \cdots \ 0]^\top. \tag{4}$$

$H_\phi \in \mathbb{R}^{N \times N}$ is the Hamiltonian of our system, and is constructed from embeddings $x$ and a bias term $H_\phi^0 \in \mathbb{R}^{N \times N}$ (Eq. 3). The bias term is a fully parametrized Hermitian matrix, giving more fine-grained control over the Hamiltonian. $U_\theta \in \mathbb{C}^{N \times N}$ represents a VQC, and $\psi_\theta$ is the result of applying said VQC to the starting state (Eq. 4). The specific choice of VQC is application-dependant and we experiment with three qubit-efficient circuits during hyperparameter tuning, which we discuss in Appendix B. The prediction probability $f_{\theta,\phi}$ is regularized using the sigmoid function $\sigma$ (Eq. 2). Finally, parameters $\theta, \phi$ are optimized classically to minimize the loss function $\mathcal{L}$, the cross entropy. When building $U_\theta$ and $\psi_\theta$, they are not explicitly represented as dense matrices in classical form. Instead, they are constructed directly on quantum hardware from a manageable set of parameters, avoiding the need for large-scale classical representations. By encoding inputs of size $d \leq N$ over $n$ qubits, we attain logarithmic scaling in the number of qubits and an overall sample complexity that scales quadratically in the embedding size. We expand on this in Section 3.6.

## 3.3 PARAMETER-EFFICIENT HAMILTONIAN (PEFF)

The bias term $H_\phi^0$ of HAM significantly contributes to its parameter count, resulting in a $O(N^2)$ scaling. To lower the model size, we experiment with applying a bias term $b_\phi \in \mathbb{R}^N$ directly to the input embeddings. This term skews the distribution in the vector space while only requiring $O(d)$ parameters. The Hamiltonian is then constructed as:

$$\tilde{x}_i := x_i + b_\phi \tag{5}$$

$$H_\phi(\tilde{x}) := \frac{1}{s} \sum_{i=1}^s \tilde{x}_i \tilde{x}_i^\top. \tag{6}$$

Physically implementing the Hamiltonians of HAM and PEFF requires first decomposing them into observables the quantum device can measure, a possibly computationally expensive process, and then performing measurements for each observable separately. In the following, we show how this can be performed, and how the model can be restructured to drastically reduce this cost.

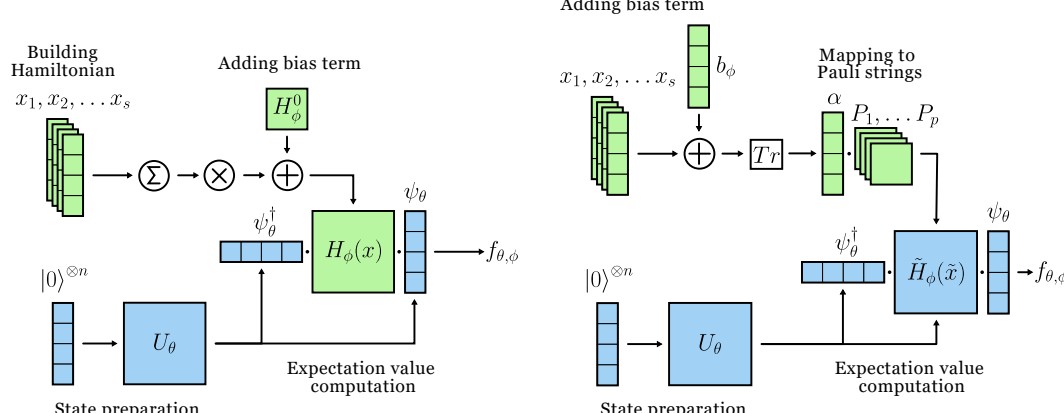

Figure 2: The HAM (left) and SIM (right) variants of our Hamiltonian Classifier at a glance. In green, parts that are stored classically, in blue, parts that can be represented on quantum computers.

## 3.4 PAULI DECOMPOSITION

The $2 \times 2$ Pauli matrices, $\mathcal{P} = \{X, Y, Z, I\}$, or more generally multi-body Pauli operators, also called *Pauli strings*, $\mathcal{P}_n = \{\bigotimes_{i=1}^{n} O_i | O_i \in \mathcal{P}\}$ form a basis for Hermitian matrices of dimension $2^n \times 2^n$. Consequently, any Hermitian matrix $H$ of size $2^n \times 2^n$ can be expressed as a linear combination of Pauli strings:

$$H = \sum_i \alpha_i P_i \quad \text{where} \quad \alpha_i \in \mathbb{R} \quad \text{and} \quad P_i \in \mathcal{P}_n.$$

This decomposition results in $O(4^n)$ unique Pauli strings, each representing a physical property that can be measured using quantum devices. In VQE settings, typical Hamiltonians consist of a polynomial number of Pauli strings. We take inspiration from this fact to rework our model.

## 3.5 SIMPLIFIED HAMILTONIAN (SIM) VARIANT

To address the computational challenges we describe above, we propose an extension of PEFF that constructs the Hamiltonian from a small number $p$ of Pauli strings, circumventing the need for an expensive decomposition and reducing the number of necessary measurements to $O(p)$ (Figure 2). We define $p$ Pauli strings $P_1, P_2, \ldots P_p \in \mathcal{P}_n$ (in practice, they can be chosen at random) and use them to compute the corresponding coefficients $\alpha_1, \alpha_2, \ldots \alpha_p$ from $H$:

$$\tilde{x} := \frac{1}{s} \sum_{i=1}^{s} x_i + b_\phi \tag{7}$$

$$H_\phi(\tilde{x}) := \tilde{x}\tilde{x}^\top \tag{8}$$

$$\alpha_i = \frac{1}{2^n} \text{Tr}(P_i H_\phi) = \frac{1}{2^n} \tilde{x}^\top P_i \tilde{x} \tag{9}$$

$$\tilde{H}_\phi(\tilde{x}) := \sum_{j=1}^{p} \alpha_j w_j P_j, \tag{10}$$

where $w_j \in \mathbb{R}$ is a learned parameter that re-weights the effect of Pauli strings. $P_j$ and $H_\phi$ in Eq. 9 are generally large matrices, but several algorithms exist that side-step the costs of full multiplication by exploiting the structure of the Pauli string (Koska et al., 2024; Hantzko et al., 2024). In this paper, we choose to redefine $H_\phi$ (Eq. 7 and 8) in a way that allows replacing the expensive matrix-matrix product $P_i H_\phi$ with two more efficient vector-matrix products $\tilde{x}^\top P_i \tilde{x}$, thus improving scaling. A related approach is discussed in Huang & Rebentrost (2024), which focuses on enhancing variational strategies rather than directly addressing the challenges of input encoding. We believe these two methods could be synergistically combined to further reduce overall computational costs.

Table 1: Theoretical scaling comparison of various VQCs, with $d$ input size, $l$ number of layers, and $p$ number of Pauli strings to measure (defined as a hyperparameter).

| Model | Reference | Qubit count | Gate complexity | Sample complexity |
|---|---|---|---|---|
| QCNN | Henderson et al. (2020) | $O(d)$ | $O(d^2)$ | $O(2^d)$ |
| | Cong et al. (2019) | $O(\log d)$ | $O(d + l \log d)$ | $O(p)$ |
| QLSTM | Chen et al. (2022) | $O(d)$ | $O(ld)$ | $O(d)$ |
| QNN | Farhi & Neven (2018) | $O(d)$ | $O(l)$ | $O(1)$ |
| | Mitarai et al. (2018) | $O(d)$ | $O(ld)$ | $O(1)$ |
| | Schuld et al. (2020) | $O(\log d)$ | $O(d + l \log d)$ | $O(1)$ |
| HAM | Ours | $O(\log d)$ | $O(l \log d)/O(l \log(d)^2)$ | $O(d^2)$ |
| PEFF | Ours | $O(\log d)$ | $O(l \log d)/O(l \log(d)^2)$ | $O(d^2)$ |
| SIM | Ours | $O(\log d)$ | $O(l \log d)/O(l \log(d)^2)$ | $O(p)$ |

To emphasize the underlying transformation, we recast the expectation value computation in SIM:

$$f_{\theta,\phi}(\tilde{x}) = \sigma\Big(\frac{1}{2^n} \sum_{j=1}^{p} \tilde{x}^\dagger P_j \tilde{x} w_j \psi_\theta^\dagger P_j \psi_\theta\Big). \tag{11}$$

Equation 11 shows that SIM computes a weighted sum of several feature maps $\tilde{x}^\dagger P_j \tilde{x}$, where the weights are determined by both the learned term $w_j$ and the factor $\psi_\theta^\dagger P_j \psi_\theta$. These two terms concur to select the feature maps most relevant for solving the problem. During our exploration, we observe that removing $w_j$ significantly degrades performance. We speculate this occurs because $\psi_\theta^\dagger P_j \psi_\theta$ is constrained to the range $[-1, 1]$, whereas the presence of $w_j$ introduces a notion of magnitude that facilitates training.

All proposed methods output a prediction probability $f_{\theta,\phi}$ to be interpreted as a binary label. We can extend this to a scenario with $c$ distinct classes by using a one-vs-many approach: either $U_\theta$, $b_\phi$ or $w$ can be tied to a specific class to obtain a class-specific discriminator. We choose a setup that learns $c$ distinct re-weightings $w_1, w_2, \ldots w_c$ and build $c$ separate Hamiltonians $\tilde{H}_\phi^1, \tilde{H}_\phi^2, \ldots \tilde{H}_\phi^c$ so that each one discriminates a single class:

$$\arg\min_{\theta,\phi} \frac{1}{|\mathbf{X}|} \sum_{x\in\mathbf{X}, y\in\mathbf{y}} \sum_{k=1}^{c} \mathcal{L}(f_{\theta,\phi}^k(x), y) \tag{12}$$

$$f_{\theta,\phi}^k(x) := \sigma(\psi_\theta^\dagger \tilde{H}_\phi^k(x)\psi_\theta) \tag{13}$$

$$\tilde{H}_\phi^k(\tilde{x}) := \sum_{j=1}^{p} \alpha_j w_j^k P_j. \tag{14}$$

Parameter count scales as $O(c)$, although different choices of parameter sharing strongly affect the final number. Since our setup learns different weights for the same Pauli strings across classes, expectation values on a real quantum device can be computed only once and then post-processed to obtain probabilities for all classes.

### 3.6 Complexity analysis

In this section, we compare the theoretical qubit count, gate complexity, and sample complexity of our classifier with other established models from the literature. We consider a subset of implementations from the literature that we consider representative of the current discourse. Our three variants all achieve a qubit count that scales as the logarithm of the input dimensionality. This is determined by the number of qubits required to encode a large enough Hamiltonian. For our models, gate complexity depends entirely on the chosen circuit $U_\theta$. The ansätze we consider throughout our experiments result in a linear or quadratic scaling. Sample complexity, defined as the total number of Pauli strings to measure in order to obtain a prediction, both HAM and PEFF necessitate a full evaluation of the Hamiltonian, resulting in a complexity of $O(4^n)$. Since $4^{\log_2(d)} = 2^{\log_2(d^2)}$, we conclude that the sample complexity is $O(d^2)$. Notably, SIM combines the logarithmic scaling in

qubit and gate complexity of the other variants with a constant sample complexity made possible by simplifying the Hamiltonian, making it a strong candidate for practical implementation on NISQ devices. The full comparison is shown in Table 1. For readability, we omit a discussion on precision when running these methods on quantum hardware. All our models incur an additional $1/\epsilon^2$ term in sample complexity, where $\epsilon$ is the desired precision. We acknowledge that this discussion is limited, as it overlooks inductive biases and other factors not captured by scaling alone. Nonetheless, we provide this comparison as a useful reference for understanding the computational trade-offs of different quantum models.

## 4 EXPERIMENTS

### 4.1 DATASETS & PRE-PROCESSING

To evaluate the capabilities of our models, we select a diverse set of tasks encompassing both text and image data, covering binary and multi-class scenarios. To facilitate replicability, our scripts automatically download all datasets on the first execution.

**Text datasets** We first consider the GLUE Stanford Sentiment Treebank (SST) dataset as obtained from HuggingFace[1] (Socher et al., 2013; Wang et al., 2019). It consists of $\sim 70k$ single sentences extracted from movie reviews whose sentiment (positive/negative) was annotated by 3 human judges. We also evaluate our method on the IMDb Large Movie Review Dataset (Maas et al., 2011) containing $50k$ highly polar movie reviews evenly split into training and test sets. Additionally, we also consider the AG News (Zhang et al., 2015) classification task as a benchmark for our multi-class model. AG News consists of $\sim 128k$ news articles divided by topic (world, sports, business, sci/tech). These are commonplace datasets reasonably close to real-world applications. Our methods and baselines all require inputs to be converted to vector representations. For text datasets, we remove all punctuation, lowercase all text, tokenize with a whitespace strategy, and finally embed tokens with word2vec[2] to obtain a sequence $x$. The resulting embedding has size $d = 300$ and, therefore, requires $n = 9$ qubits to be represented in our methods.

**Image datasets** As a sanity check, we consider a binary version of MNIST which only includes digits 0 and 1. We then consider Fashion-MNIST (Xiao et al., 2017), a dataset of $60k$ grayscale images of clothes subdivided into 10 classes. Feeding images directly to our models results in $d = 784$ and $n = 10$. We further experiment on a binarized version of the CIFAR-10 dataset (Krizhevsky, 2009) we name CIFAR-2 obtained by grouping the original ten classes into two categories: vehicles and animals. The $32 \times 32$ RGB images result in $d = 3 \times 1024$ features over $n = 12$ qubits. We also note that no further pre-processing or dimensionality reduction is performed to preserve the original properties of the data. MNIST and Fashion-MNIST representations are not sequential and therefore are considered by our model as a special case with $s = 1$, while in CIFAR-2 each channel is considered as a different element of a sequence with $s = 3$.

### 4.2 BASELINES

We compare our classifiers (HAM, PEFF, SIM) with three quantum baselines: a QLSTM (Chen et al., 2022), a QCNN (Cong et al., 2019), and a simple circuit ansatz (CIRC). QLSTM and QCNN have been adapted from implementations of the original papers. CIRC is our implementation and consists of an amplitude encoding for the input, the same hardware-efficient ansätze of HAM, and a linear regression on the circuit's output state. Note that running CIRC in practice would require state tomography, this setup is therefore not meant to be efficient or scalable but rather to give a best-case scenario of a VQC of similar complexity to our classifier. We also compare with out-of-the-box classical baselines: a multi-layer perception (MLP), a logistic regression (LOG), an RNN, an LSTM, and a CNN. MLP and LOG act on the mean-pooled embedding of the sequence.

---

[1]https://huggingface.co/datasets/stanfordnlp/sst2
[2]https://code.google.com/archive/p/word2vec/

Table 2: Accuracies across text datasets. Results are averaged over 10 runs or (*) run achieving lowest training loss.

| Model | SST2 | | | IMDb | | | AG News | | |
|---|---|---|---|---|---|---|---|---|---|
| | # Params | Train Acc | Test Acc* | # Params | Train Acc | Test Acc | # Params | Train Acc | Test Acc |
| LOG | 301 | 84.7 | 80.4 | 301 | 85.8 | 85.5 | 1204 | 90.1 | 89.2 |
| MLP | 180901 | 97.5 | 80.2 | 180901 | 88.1 | 85.8 | 40604 | 90.2 | 89.1 |
| LSTM | 241701 | 97.8 | 84.4 | 241600 | 99.3 | 88.4 | 242004 | 91.5 | 90.5 |
| RNN | 40301 | 89.6 | 80.1 | 60501 | 79.7 | 78.1 | 40604 | 88.8 | 88.1 |
| CIRC | 923 | 85.2 | 80.1 | 923 | 86.1 | 85.8 | 2387 | 89.7 | 89.2 |
| QLSTM | 2766 | 89.9 | 84.4 | 4679 | 76.8 | 67.9 | 3582 | 87.2 | 86.7 |
| HAM | 130854 | 91.2 | 82.3 | 130926 | 91.0 | 88.1 | - | - | - |
| PEFF | 410 | 81.8 | 80.0 | 410 | 84.0 | 83.8 | - | - | - |
| SIM | 1338 | 80.6 | 79.0 | 1410 | 86.0 | 85.3 | 4416 | 89.4 | 88.6 |

Table 3: Accuracies across image datasets. Results are averaged over 10 runs.

| Model | MNIST2 | | | CIFAR2 | | | Fashion | | |
|---|---|---|---|---|---|---|---|---|---|
| | # Params | Train Acc | Test Acc | # Params | Train Acc | Test Acc | # Params | Train Acc | Test Acc |
| LOG | 785 | 100.0 | 100.0 | 1025 | 73.9 | 72.9 | 1025 | 85.5 | 83.5 |
| MLP | 88701 | 100.0 | 99.9 | 112701 | 89.0 | 83.4 | 89610 | 87.9 | 85.8 |
| CNN | 26065 | 100.0 | 100.0 | 75329 | 96.3 | 94.1 | 130250 | 95.4 | 90.7 |
| CIRC | 1841 | 99.9 | 99.9 | 2081 | 85.1 | 84.6 | 11094 | 89.2 | 86.5 |
| QCNN | 169 | 99.8 | 99.8 | 169 | 84.9 | 84.7 | 558 | 76.7 | 76.3 |
| HAM | 523808 | 100.0 | 99.9 | 523818 | 89.6 | 81.0 | - | - | - |
| PEFF | 916 | 99.9 | 99.9 | 1056 | 79.3 | 78.7 | - | - | - |
| SIM | 1826 | 100.0 | 99.9 | 2156 | 68.9 | 68.2 | 10934 | 87.6 | 84.4 |

## 4.3 EXPERIMENTAL SETTING

We first perform a random search on hyperparameters such as learning rate, batch size, number of qubits, and number of layers to identify the best configuration of each architecture. Because of space constraints, a more detailed discussion is moved to Appendix B. After identifying the best hyperparameters, we train 10 models for each architecture with randomized seeds on the original training split and average their performance on the test split. The only exception is the SST2 dataset for which no test set is provided. In this case, we select the run achieving the lowest training loss and submit it to GLUE's website to get a test score. Submitting all 10 runs for each model would require an unfeasible amount of time given the maximum 2 daily submissions imposed by the platform. Appendix C shows additional experiments in which we further investigate architectural choices such as bias, state preparation, and number of Pauli strings. With the exception of QCNN which is implemented in PennyLane, all quantum operations are simulated in PyTorch as it allows batching several Hamiltonians thus enabling efficient training. We do not simulate noise in order to assess the performance of our approach under an ideal scenario. For all tasks we use a cross-entropy loss during training.

## 4.4 RESULTS

In Tables 2 and 3, we report train and test accuracy for the text and image datasets respectively. All models successfully achieve perfect scores in the MNIST2 setup, validating our setup and confirming that HAM, PEFF, and SIM learn correctly. In the SST2 task, HAM performs competitively with baseline methods using a similar number of parameters, outperforming simpler models like LOG, MLP and RNN, and ranking just below the LSTM and QLSTM. AG News and IMDb turn out to be easier tasks, with SIM achieving scores comparable to the baselines and accuracy being high for all models. CIFAR2 proves to be a hard task: while HAM and PEFF score relatively higher than the LOG baseline, SIM underperforms possibly due to its relatively simple decision boundary. On Fashion, SIM performs better than LOG but worse than other baselines. Notably, CIRC achieves

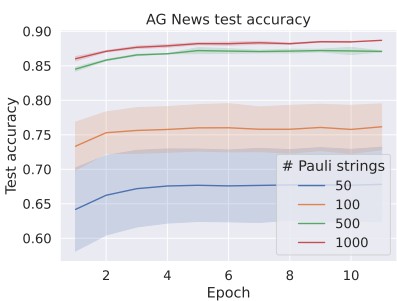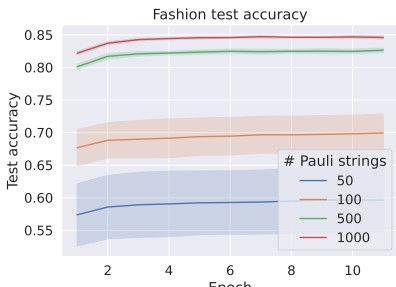

Figure 3: Performance on the test sets for different number of Pauli strings in the SIM model. First 10 epochs out of 30 shown.

relatively high performance. Across tasks, PEFF and SIM attain performance comparable with the baseline despite the low parameter count. PEFF confirms our intuition that simply skewing points in the embedding space can substitute the bulky bias term over the whole Hamiltonian, while SIM confirms that performance can be retained even with a Hamiltonian composed of few Pauli strings. The high parameter count of HAM in MNIST2 is necessary to encode the full $28 \times 28$ pixel images, resulting in large Hamiltonians.

Across quantum models, we find no clear link between entangling circuits and performance; non-entangling baselines often perform best, aligning with prior findings (Bowles et al., 2024). QLSTM and QCNN struggle to learn and display a less stable hyperparameter tuning, making it challenging to find well-performing configurations.

We perform additional experiments in Appendix C and find evidence that the number of Pauli strings is strongly linked with better performance. Specifically, larger models with more Pauli strings exhibit higher accuracy and more stable training dynamics (Figure 3). Notably, between 500 and 1000 Pauli strings are already sufficient to match the performance of classical baselines on most tasks. However, for more complex datasets like CIFAR2, where performance currently lags, we believe increasing the number of Pauli strings beyond 1000 could significantly improve results. This is suggested by Eq. 11, which shows how increasing the number of Pauli strings enables the model to capture more complex features. Furthermore, we find that removing the bias term significantly worsens performance, underlining the importance of this component.

## 5 DISCUSSION AND FUTURE WORK

This works highlights how measurement can become a central part of quantum computation while at the same time alleviating the costs of data manipulation on quantum devices. Our Hamiltonian classifier achieves promising results on several domains, competing with classical baselines. It demonstrates how alternative encoding strategies can bypass some hardware limitations while scaling sufficiently well for real-world problems. The proposed HAM design encodes input data directly as a measurement, obtaining performance comparable with other specialized models. The PEFF variant reduces its parameter complexity, and the SIM variant additionally reduces its sample complexity, offering a scheme that may be more efficiently implemented on quantum hardware. Notably, our method already scales well enough to allow meaningful studies on large datasets using simulators. The Hamiltonian classifier is presently a proof of concept meant to illustrate a novel input scheme for quantum devices with the ultimate goal of expanding the tools available to QML researchers. Future works could characterize the effectiveness of our approach on even larger problems and its integration with existing classical pipelines. Other studies could focus on how different choices of Pauli strings affect the final outcome. For example, exploring ways to integrate inductive biases tailored to specific tasks, or consider local strings in conjunction with classical shadow techniques to lower sample complexity. A way of learning more complex functions could be to stack several layers of Hamiltonian encoding, possibly performing nonlinear transformations. Other directions deserving a paper of their own are noise simulation and physical implementations on real quantum hardware.

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

## A QUANTUM COMPUTING FUNDAMENTALS

Quantum computers are conceptually and physically different devices from classical computers. Instead of basing their logic on binary data representations implemented by transistors, they utilize the rules of quantum physics as a substrate for computation. For this reason, in what follows we describe the building blocks of quantum computation.

**Dirac notation** Quantum computing makes extensive use of Dirac notation (also called bra-ket notation) to simplify the representation of linear transformations which are commonplace in the theory. The fundamental elements of this notation are *bras* and *kets*. A ket, denoted as $|\psi\rangle$, represents a column vector $\psi$ in a Hilbert space. A bra, denoted as $\langle\phi|$, represents instead a vector in the dual space (the complex conjugate transpose of a ket). The inner product of two vectors $\phi$ and $\psi$, which results in a scalar, is written as $\langle\phi|\psi\rangle$. Conversely, the outer product $|\psi\rangle\langle\phi|$, forms a matrix or an operator.

**Quantum systems** The basic unit of information in a quantum computer is a two-dimensional quantum bit (qubit) which is mathematically modelled by a normalized column vector, the *state vector*, in a two-dimensional vector space equipped with an inner-product, or Hilbert space. A state vector is usually expressed concisely in Dirac notation as $|\psi\rangle = [\alpha \; \beta]^\top \in \mathbb{C}^2$. We say that a qubit $|\psi\rangle := \alpha |0\rangle + \beta |1\rangle$ is in a *coherent quantum superposition* of two orthonormal basis states $|0\rangle := [1 \; 0]^\top$ and $|1\rangle := [0 \; 1]^\top$, such that the complex coefficients in the basis expansion satisfy normalization constraint $|\alpha|^2 + |\beta|^2 = 1$. This constraint originates from physics: in order to extract any information from a quantum state, we need to *measure* it in a chosen basis, which destroys the superposition in the measured basis by projecting the state $|\psi\rangle$ on one of the basis elements. If we measure state $|\psi\rangle$ in the $\{|0\rangle, |1\rangle\}$ basis, we get outcome "0" with probability $|\alpha|^2$ and the post-measurement state becomes $|0\rangle$, and respectively, we get "1" with probability $|\beta|^2$ and post-measurement state $|1\rangle$.

**Multi-qubit systems** In order to model a quantum system with multiple qubits, we use the so-called *Kronecker product* ($\otimes$) to combine many individual state vectors into a single larger one. An $n$-qubit system can be represented by a vector of size $N = 2^n$, $|\psi_1 \ldots \psi_n\rangle := |\psi_1\rangle \otimes \cdots \otimes |\psi_n\rangle$. In a chosen basis, the entries of this vector describe the probability of observing that outcome. Many QML approaches aim to gain a quantum advantage by manipulating only a few qubits to access an exponentially large Hilbert space. To give an example, if two qubits $|\psi_1\rangle$ and $|\psi_2\rangle$ are both initialized in $(|0\rangle + |1\rangle)/\sqrt{2}$, the joint state is given by an equal superposition of all the possible $n = 2$ bit string vectors

$$|\psi_1\rangle \otimes |\psi_2\rangle = \frac{|00\rangle + |01\rangle + |10\rangle + |11\rangle}{2}.$$

In this setup, measurement can performed separately on each qubit, resulting in a $n$-element bit-string.

**Quantum circuits** Quantum computation is achieved by manipulating qubits. This is done using quantum *gates*. Any $n$-qubit gate can be represented as a unitary matrix $U \in \mathbb{C}^{N \times N}$ which acts on the $n$-qubit input state $|\psi\rangle$ via the usual matrix-vector multiplication, giving output $U |\psi\rangle$. Intuitively, quantum gates can be considered as rotations that conserve the length of a state vector. A sequence of gates applied on one or many qubits is called a quantum *circuit*. By construction, unitary circuits perform linear operations. Non-linear computation requires workarounds like running the computation on a larger Hilbert space and measuring output qubits in a subspace or re-uploading input data (Killoran et al., 2019; Pérez-Salinas et al., 2020; Subasi et al., 2023).

# B  HYPERPARAMETER TUNING

For each architecture and for each dataset in our evaluation, we perform a randomised search for the best parameters: we randomly select 50 configurations, train them on the task and evaluate their performance on the development set. To avoid overfitting, we perform early stopping on the training if development loss does not decrease for five consecutive epochs. Troughout all experiments, we utilize the *Adam* optimizer provided by PyTorch. Since our largest model HAM has at most $\sim 500k$ parameters (dictated by the embedding space), we limit the random search to configuration with less-than or equal number of parameters to ensure a fair comparison. In practice, we find RNNs, LSTMs and CNNs perform very well with as low as $40k$ parameters. What follows is a list of all hyperparameters we evaluated:

- **Batch size:** $[64, 128, 256]$;
- **Learning rate:** $[10^{-2}, 10^{-3}, 10^{-4}]$;
- **Hidden size:** For RNNs and LSTMs, the size of the hidden representation $[100, 300, 500]$;
- **Layers:** For RNNs and LSTMs, the number of layers of the recurrent block $[1, 4, 8]$. For MLP, the total number of layers including input and output $[3, 4, 5]$. For CNNs, the total number of convolutional layers $[3, 4, 5]$. For HAM, PEFF, SIM and CIRC, the number of repeated applications of the ansatz, analogous to the number of layers $[8, 16, 32]$;
- **Channels:** For CNNs, the number of output channels of each convolutional layer $[8, 16, 32, 64, 128]$;

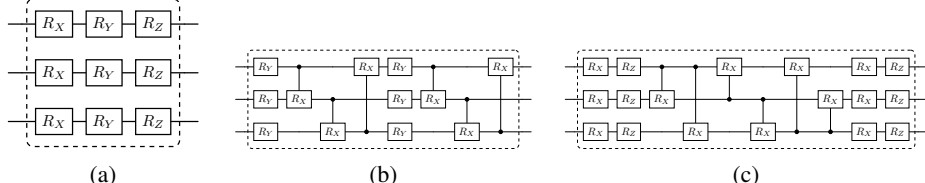

(a)  (b)  (c)

Figure 4: Circuit ansätze explored in the experiment. For the sake of visualization, the image shows ansätze for $n = 3$. Since the experiments use different values, we extend the patterns to act on more qubits. (a) Non-entangling ansatz (b) Ring ansatz (c) All-to-all ansatz. (b) and (c) are respectively circuit 14 and 6 from (Sim et al., 2019)

- **Kernel size:** For CNNs, the kernel dimension of each convolutional layer $[3, 5, 7]$;

- **Circuit:** The ansatz $U_\theta$ used to prepare state $\psi$. We experiment with three hardware-efficient ansätze inspired by circuits from Sim et al. (2019) and illustrated in Figure 4. These ansätze are designed to explore different levels of entanglement (the quantum counterpart of correlation) and gate complexity. The first is a non-entangling circuit composed of single-qubit rotations, exhibiting linear complexity with respect to the number of qubits. The second is an entangling circuit arranged in a ring configuration, also scaling linearly in gate count. The third is an all-to-all entangling circuit, which scales quadratically. [Baseline, Ring, All-to-all].

- **# Pauli strings:** For SIM, the number of Pauli strings composing the Hamiltonian $[50, 100, 500, 1000]$.

Tables 4 and 5 shows the final selection of best-performing hyperparameters used in the main experiment of Section 4.3.

## C  SUPPLEMENTARY EXPERIMENTS

### C.1  A LEARNED BIAS BOOSTS PERFORMANCE

To further evaluate our models and assess the impact of key design choices, we conduct an ablation study using the SST2 dataset. As in the main experiment, we begin with the optimal set of hyperparameters identified in Appendix B, apply changes to the architecture, and run 10 separate runs. Finally, we report the test set performance of the run achieving the lowest loss as reported on the GLUE website.

In this ablation study, we examine the role of the bias term in contributing to downstream performance. Our best-performing model, HAM, delivers strong results but does so with a large number of parameters. In contrast, PEFF offers slightly lower performance but significantly reduces the parameter count, as detailed in Section 4.4. To explore the impact of the bias term we completely remove it and assess the performance of the resulting model, referring to this variant as NOBIAS:

$$H_\phi(x) \coloneqq \frac{1}{s} \sum_{i=1}^{s} x_i x_i^\top. \tag{15}$$

The results, summarized in Table 6, show a marked decrease in performance. The $71.9\%$ accuracy achieved by NOBIAS is nonetheless surprising given the model is stripped of almost all parameters. We hypothesize that, in HAM, the bias term influences the eigenvalues of the Hamiltonian, directly affecting the final expectation value. For PEFF, the bias appears to shift the embeddings into a region more favorable for classification. We further speculate that NOBIAS could achieve results similar to PEFF by simply unfreezing the input embeddings during training, this would allow them to shift as in the other variants.

Table 4: Best hyperparameters across all baseline models and tasks

**LOG**

| Dataset | Batch size | Learning rate |
|---------|-----------|---------------|
| SST2 | 64 | $10^{-3}$ |
| IMDb | 64 | $10^{-2}$ |
| AG News | 64 | $10^{-2}$ |
| MNIST2 | 128 | $10^{-3}$ |
| CIFAR2 | 256 | $10^{-3}$ |
| Fashion | 64 | $10^{-3}$ |

**MLP**

| Dataset | Batch size | Learning rate | Hidden size | Layers |
|---------|-----------|---------------|-------------|--------|
| SST2 | 64 | $10^{-4}$ | 300 | 3 |
| IMDb | 64 | $10^{-3}$ | 300 | 3 |
| AG News | 64 | $10^{-3}$ | 100 | 3 |
| MNIST2 | 256 | $10^{-4}$ | 100 | 3 |
| CIFAR2 | 128 | $10^{-4}$ | 100 | 3 |
| Fashion | 256 | $10^{-3}$ | 100 | 3 |

**LSTM**

| Dataset | Batch size | Learning rate | Hidden size | Layers |
|---------|-----------|---------------|-------------|--------|
| SST2 | 64 | $10^{-2}$ | 100 | 2 |
| IMDb | 128 | $10^{-2}$ | 100 | 2 |
| AG News | 128 | $10^{-3}$ | 100 | 2 |

**RNN**

| Dataset | Batch size | Learning rate | Hidden size | Layers |
|---------|-----------|---------------|-------------|--------|
| SST2 | 64 | $10^{-4}$ | 100 | 1 |
| IMDb | 256 | $10^{-4}$ | 100 | 2 |
| AG News | 256 | $10^{-4}$ | 100 | 1 |

**CNN**

| Dataset | Batch size | Learning rate | Layers | Channels | Kernel Size |
|---------|-----------|---------------|--------|----------|-------------|
| MNIST2 | 64 | $10^{-3}$ | 5 | 16 | 5 |
| CIFAR2 | 64 | $10^{-4}$ | 4 | 32 | 5 |
| Fashion | 64 | $10^{-3}$ | 4 | 16 | 3 |

**CIRC**

| Dataset | Batch size | Learning rate | Circuit | Layers |
|---------|-----------|---------------|---------|--------|
| SST2 | 256 | $10^{-3}$ | All-to-all | 8 |
| AG News | 128 | $10^{-3}$ | Baseline | 16 |
| IMDb | 128 | $10^{-3}$ | All-to-all | 16 |
| MNIST2 | 128 | $10^{-3}$ | Baseline | 8 |
| CIFAR2 | 64 | $10^{-3}$ | Baseline | 8 |
| Fashion | 256 | $10^{-3}$ | Ring | 32 |

Table 5: Best hyperparameters across all Hamiltonian models and tasks

| HAM | | | | |
| Dataset | Batch size | Learning rate | Circuit | Layers |
| --- | --- | --- | --- | --- |
| SST2 | 128 | $10^{-3}$ | Ring | 8 |
| IMDb | 256 | $10^{-3}$ | All-to-all | 32 |
| MNIST2 | 256 | $10^{-2}$ | Baseline | 32 |
| CIFAR2 | 64 | $10^{-3}$ | Ring | 8 |

| PEFF | | | | |
| Dataset | Batch size | Learning rate | Circuit | Layers |
| --- | --- | --- | --- | --- |
| SST2 | 64 | $10^{-2}$ | All-to-all | 8 |
| IMDb | 64 | $10^{-2}$ | All-to-all | 8 |
| MNIST2 | 64 | $10^{-3}$ | All-to-all | 8 |
| CIFAR2 | 256 | $10^{-2}$ | Baseline | 16 |

| SIM | | | | | |
| Dataset | Batch size | Learning rate | Circuit | Layers | # Pauli strings |
| --- | --- | --- | --- | --- | --- |
| SST2 | 256 | $10^{-3}$ | Ring | 32 | 1000 |
| IMDb | 256 | $10^{-2}$ | All-to-all | 16 | 1000 |
| AG News | 128 | $10^{-3}$ | All-to-all | 16 | 1000 |
| MNIST2 | 256 | $10^{-2}$ | Ring | 32 | 1000 |
| CIFAR2 | 64 | $10^{-3}$ | All-to-all | 32 | 1000 |
| Fashion | 256 | $10^{-2}$ | All-to-all | 8 | 1000 |

Table 6: Accuracies of additional variants on SST2. Results are averaged over 10 runs or (*) run achieving best train accuracy.

| | # Params | Train Acc | Test Acc* |
| --- | --- | --- | --- |
| HAM | 130854 | 91.2 | 82.3 |
| PEFF | 410 | 81.8 | 80.0 |
| SIM | 9410 | 84.5 | 80.1 |
| STATEIN | 130854 | 92.0 | 80.2 |
| NOBIAS | 38 | 70.0 | 71.9 |

## C.2 ADDITIONAL STATE PREPARATION LEADS TO OVERFITTING

In this section we investigate whether combining our HAM architecture with a standard state preparation routine improves model performance. Also in this experiment we consider SST2, with the configuration of the main text unchanged but for the additional state preparation. We call this setup STATEIN (short for STATE INput):

$$\tilde{x} := \frac{1}{s} \sum_{i=1}^{s} \tilde{x}_i \tag{16}$$

$$\psi_\theta := U_\theta \left| \tilde{x} \right\rangle \tag{17}$$

In this configuration, $U_\theta$ acts not on the zero-state, but on an initial state $\left| \tilde{x} \right\rangle$ in which the input data has been amplitude-encoded. We hypothesize that applying $U_\theta$ directly to the encoded inputs might enable the model to better identify useful features for classification. Results displayed in Table 6 show a slight improvement in training accuracy, with STATEIN achieving an average of 92.0% over 10 runs compared to 91.1% for HAM. However, the test accuracy only marginally exceeds that of PEFF and is noticeably lower than HAM, suggesting that the combination of Hamiltonian encoding and amplitude-encoded inputs may lead to overfitting. While these results provide some insights, a more thorough analysis of the interaction between state preparation and input-encoded measurement is needed, which we leave for future research.

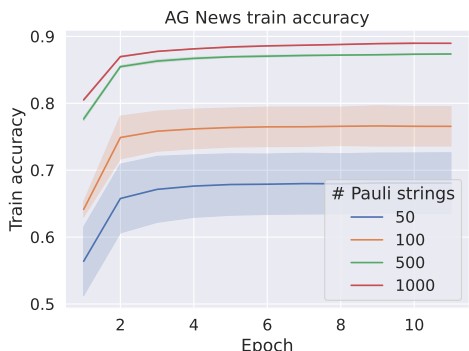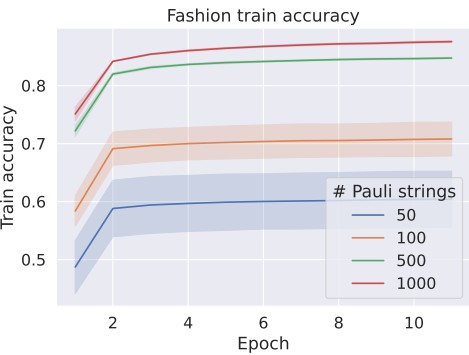

Figure 5: Performance on the train set for different number of Pauli strings in the SIM model. Error bars are shown for all choices but grow thin for the two largest models. First 10 epochs out of 30 shown.

### C.3 THE NUMBER OF PAULI STRINGS CORRELATES WITH PERFORMANCE

Rewriting the SIM model leads to a bilinear form where each Pauli string acts as a transformation on the input (Eq. 11). During hyperparameter tuning, we consistently observe that the best-performing SIM models use 1000 Pauli strings across all experiments. We believe this is not a coincidence, as each Pauli string introduces a distinct transformation, enriching the model's feature set. This raises some questions: does increasing the number of Pauli strings, and thus the number of transformations, lead to better performance? Moreover, how does the generalization capability scale with the number of Pauli strings?

To explore this, we use the same configurations found for the AG News and Fashion datasets but vary the number of Pauli strings. Results are displayed in Figures 3 and 5. Low string count do not perform much better than chance, but accuracy steadily increases and eventually plateaus at 1000 strings, reaching levels comparable to other models like MLP and CIRC. Increasing the number of strings also leads to a more stable training process as highlighted by the error bars growing thin for $p = 500$ and 1000. This suggests the loss landscape may become smoother as more transformations are added, facilitating training. This is a promising outcome: it suggests that transformations induced by Pauli strings actively contribute to learning by creating more complex features. It also indicates that a relatively small number of strings can effectively substitute for a full decomposition.

