# OpenReview forum: "An Efficient Quantum Classifier Based on Hamiltonian Representations"
_ICLR.cc/2025/Conference — ICLR 2025 Conference Withdrawn Submission_

### Official Review · Reviewer_uKu1 · 2024-10-23

**Soundness:** 3
**Presentation:** 2
**Contribution:** 1
**Rating:** 3
**Confidence:** 5

**Summary:**

This paper argues that previous NISQ quantum machine learning algorithms generally require a large number of computational resources, which may not be suitable for NISQ devices. To address the data-encoding problem, the authors encode classical data into a physical Hamiltonian, referred to as the Hamiltonian classifier. To demonstrate the power of the Hamiltonian classifier, they numerically test their model on several popular classical datasets, including text and image classification tasks.

**Strengths:**

From a high-level perspective, this paper is well-written, clearly introducing the research background and presenting their results. Meanwhile, the authors have put significant effort into the numerical simulation section, where they compare various related quantum-classical methods across several datasets.

**Weaknesses:**

(1) One of the main concerns is that the "Hamiltonian Classifier" has been proposed and studied in several papers, such as [S. Jerbi et al., Shadows of Quantum Machine Learning, Nat. Comm., 2024] and [Y. Song et al., A Quantum Federated Learning Framework for Classical Clients, Sci. China Phys. Mech. Astron. (2024)]. However, this paper does not cite these highly relevant works. As a result, the authors' contributions are significantly diminished, especially the claim in Sec. 3, page 4, "To the best of our knowledge..."

(2) As a high-level conference (ICLR) in the field of machine learning, we expect to see more surprising results in quantum machine learning. The authors still utilize the standard classical-quantum workflow, despite encoding the data into a physical Hamiltonian. While many papers adopt this framework and numerically benchmark their methods on some datasets, such research contributes very little to quantum computation theory and the quantum machine learning community, particularly at this stage. This is supported by recent findings: it has been shown that many classical-quantum QML methods (including the authors') are classically simulable when the model does not suffer from the barren plateaus phenomenon [A. Angrisani et al., Classically Estimating Observables of Noiseless Quantum Circuits, arXiv:2409.01706]. Furthermore, when the QML algorithm is limited to a 2D architecture, all constant-depth (or constant evolution time) quantum-classical approaches can be classically simulated [S. Bravyi et al., Classical Algorithms for Quantum Mean Values, Nat. Phys., 2021]. From this perspective, it appears that the Hamiltonian classifier method may not provide a clear quantum advantage.

**Questions:**

Here are some minor questions:
(1) On Page 6, authors claimed that they can randomly define $p$ Pauli strings, and decompose the data matrix $H_{\phi}(\tilde{x})$ onto the sampled Pauli basis. Then, is there any creteria on selecting these Pauli strings? What is the scaling of the parameter $p$, and what is the relationship between $p$ and the power of Hamiltonain classifier model (such as generalization error upper bound or the effective dimension)?
(2) In equation 10, $\alpha_i$ has an exponentially small factor $2^{-n}$. Does this factor cause the measurement accuracy to become exponentially small, leading to a large amount of measurement overhead?
(3) In Table 1, Cong et al. (2019) do not utilize the QCNN to solve a classical task; instead, they predict the quantum phase transition problem. Is it fair to compare Cong et al. (2019) to the authors' work on a classical task?
(4) In Tables 2 and 3, it is observed that the proposed methods (HAM, PEFF, and SIM) do not outperform all the listed methods. Given these facts, what is the advantage of the proposed Hamiltonian classifier method?

---

> ### Author Response · Authors · 2024-11-20
>
> We thank the reviewer for their positive feedback. We appreciate the acknowledgement of the paper's clarity as well as the value of our experimental setup.
>
> We now address the concerns raised in the following sections.
>
> > (1) One of the main concerns is that the "Hamiltonian Classifier" has been proposed and studied in several papers, such as [S. Jerbi et al., Shadows of Quantum Machine Learning, Nat. Comm., 2024] and [Y. Song et al., A Quantum Federated Learning Framework for Classical Clients, Sci. China Phys. Mech. Astron. (2024)]. However, this paper does not cite these highly relevant works. As a result, the authors' contributions are significantly diminished, especially the claim in Sec. 3, page 4, "To the best of our knowledge..."
>
> We appreciate the reviewer’s suggestion to cite the relevant works by Jerbi et al. and Song et al. While we were unaware of these studies, we have since reviewed them and incorporated the citations. We also clarify that our approach applies a specific instance of the "flipped" model to our tasks.
>
> Unlike the referenced papers, which provide a general classification of flipped models, our contribution lies in the detailed evaluation of various models and datasets, highlighting the method’s applicability across different contexts.
>
> > (2) As a high-level conference (ICLR) in the field of machine learning, we expect to see more surprising results in quantum machine learning. The authors still utilize the standard classical-quantum workflow, despite encoding the data into a physical Hamiltonian. [...] From this perspective, it appears that the Hamiltonian classifier method may not provide a clear quantum advantage.
>
> We again thank the reviewer for the insightful comments. After reviewing the provided literature we decided to rephrase the message of the paper. Specifically, obtaining stronger results than classical methods is unfeasible without stronger quantum hardware (i.e. on the scale required to solve cryptographic tasks) (https://arxiv.org/abs/2208.06339,
> https://www.semanticscholar.org/paper/Equivalences-and-Separations-Between-Quantum-and-Servedio-Gortler/bdb567bb253b9f57911b267ab568c8dcc591400d), and that our method highlights how close a simple method can get to classical performance in the current NISQ regime.
>
> > (1) On Page 6, authors claimed that they can randomly define
> Pauli strings, and decompose the data matrix onto the sampled Pauli basis. Then, is there any creteria on selecting these Pauli strings? What is the scaling of the parameter , and what is the relationship between and the power of Hamiltonain classifier model (such as generalization error upper bound or the effective dimension)?
>
> Our study primarily focuses on empirical investigations rather than theoretical analyses. In Section 4.4, we present experimental results that explore the scaling behavior, and we provide further details and expanded discussions in Appendix C.
>
> > (2) In equation 10, has an exponentially small factor . Does this factor cause the measurement accuracy to become exponentially small, leading to a large amount of measurement overhead?
>
> No, this is a global factor that can be omitted without impacting performance. It was put there to match the standard Pauli decomposition formulation.
>
> > (3) In Table 1, Cong et al. (2019) do not utilize the QCNN to solve a classical task; instead, they predict the quantum phase transition problem. Is it fair to compare Cong et al. (2019) to the authors' work on a classical task?
>
> While Cong et al. (2019) apply QCNNs to a quantum phase transition problem, the use of QCNNs on classical tasks, including image classification, has been demonstrated in previous works (e.g., https://link.springer.com/article/10.1007/s10044-022-01113-z, https://arxiv.org/abs/2408.08701v1). Although adapting the QCNN for optimal performance on our specific dataset is possible, we opted for minimal changes to demonstrate its out-of-the-box applicability to classical tasks. Better performance could be achieved but was out of the scope of this work.
>
> > (4) In Tables 2 and 3, it is observed that the proposed methods (HAM, PEFF, and SIM) do not outperform all the listed methods. Given these facts, what is the advantage of the proposed Hamiltonian classifier method?
>
> The advantage of our proposed Hamiltonian classifier does not necessarily lie in achieving the highest accuracy but in significantly reducing quantum encoding costs and circuit depth. These enhancements are crucial for near-term quantum devices, as they enable the application of our method to larger-scale machine learning tasks. While further research may improve classification performance, this work serves as a demonstration of a promising approach for practical quantum machine learning.

---

> > ### Comment · Reviewer_uKu1 · 2024-11-21
> > **Response**
> >
> > Q 1: The authors have stated that the proposed method is merely an instance of previously related works. Such a contribution may not meet the acceptance standards of ICLR.
> >
> > Q 2: As I mentioned in the first round, the proposed method remains within the standard quantum-classical workflow and focuses on widely used datasets, which is significantly different from the paper [https://arxiv.org/abs/2208.06339], which features a rigorous mathematical structure. And the author's response is a little bit confused, since the advantage does not always require the 'strong hardware' (e. g. S. Bravyi et al., quantum advantages with shallow circuits). Furthermore, the proposed method can still be classically simulated using the approach described in  [A. Angrisani et al., Classically Estimating Observables of Noiseless Quantum Circuits, arXiv:2409.01706].
> >
> > Due to the very limited theoretical innovation in this paper, as the model used is still a special case of previous works and does not make a substantial contribution to the field of quantum machine learning, I do not recommend its publication at the ICLR conference.

---

### Official Review · Reviewer_T8KM · 2024-10-27

**Soundness:** 3
**Presentation:** 3
**Contribution:** 2
**Rating:** 3
**Confidence:** 3

**Summary:**

Authors provide a new method to implement variational quantum circuits (VQCs) that can be used for machine learning tasks using quantum hardware. This method achieves a logarithmic scaling in both qubits and quantum gate counts, while having worse sample complexity (as presented in Table 1). They then numerically simulate their quantum algorithm on text and image classification, and achieve results that are mostly better than other forms of VQCs and on-par with classical neural networks (such as MLPs and CNNs).

**Strengths:**

This paper is clearly written and scientifically sound. The improvement in gate complexity is substantial, since no other methods achieve logarithmic scaling for these quantities.
I believe the paper provides a good advance in the field of variational quantum circuits.

**Weaknesses:**

The main critique of this paper is that the method achieves mostly the same performance than very simple classical models, such as logistic regression, MLPs and CNNs, while being based on having access to ideal quantum hardware which is not readily available (and will only be made available in the long-term). Therefore, the main value of the paper is an algorithmic advance for VQCs (and in how data is encoded into the quantum computer), but not for the global field of machine learning itself.
Simulations that include noise could be useful, which hopefully would show that even on noisy quantum hardware the results are not significantly altered.
The benchmarks are somewhat sparse, and some of them include results that have 100% test accuracy for many methods, so we cannot know which is better. Considering more difficult datasets would be more interesting.
The maximum number of qubits considered is $n=10$ if I am not mistaken, which is significantly smaller than what can be simulated with even small computational resources ($n=20$ is feasible in the noiseless setting).

**Questions:**

In the complexity analysis, section 3.6, you mention that your method incurs an additional $1/\epsilon^2$ cost. Is this the same for other methods presented in Table 1?

---

> ### Author Response · Authors · 2024-11-20
>
> We thank the reviewer for recognizing our paper as a meaningful advance in the field of variational quantum circuits. We are also grateful for the positive remarks on the clarity and scientific soundness of the work, as well as the acknowledgment of our substantial improvement in gate complexity.
>
> We now address the reviewer’s critiques and outline the steps taken to strengthen the manuscript.
>
> >The main critique of this paper is that the method achieves mostly the same performance than very simple classical models, such as logistic regression, MLPs and CNNs, while being based on having access to ideal quantum hardware which is not readily available (and will only be made available in the long-term). Therefore, the main value of the paper is an algorithmic advance for VQCs (and in how data is encoded into the quantum computer), but not for the global field of machine learning itself.
>
> While it is true that our work primarily focuses on advancing Variational Quantum Circuits, this contribution is far from trivial. The challenge of demonstrating quantum models that outperform classical baselines remains a widely acknowledged hurdle in the field of quantum machine learning. Indeed, as benchmarks have shown (Bowles et al., 2024), matching classical performance alone can already represent a meaningful achievement.
>
> The contributions of our paper go beyond performance comparisons: by reducing circuit depth and sample complexity, we address critical scalability issues in VQCs, which are central to their practical adoption. Moreover, we showcase how these techniques can already handle sizable ML tasks, offering a path toward their broader adoption and contributing to ML at large.
>
> Finally, we note that studies revolving entirely around VQCs have been accepted to this venue in the past, demonstrating that such work is a valid and valuable contribution (https://iclr.cc/virtual/2023/poster/11285, https://iclr.cc/virtual/2023/poster/11652).
>
> > Simulations that include noise could be useful, which hopefully would show that even on noisy quantum hardware the results are not significantly altered.
>
> We acknowledge the value of noisy simulations in principle, but their implementation is challenging in our case. Existing noise simulation frameworks lack support for batch processing of Hamiltonians, a crucial feature for running our algorithm efficiently. This limitation led us to emulate a quantum computer using PyTorch, deferring noisy simulations to future work when suitable frameworks become available. Additionally, without strict theoretical guarantees for the algorithm, noisy simulations may not provide actionable insights at this stage. We believe addressing these theoretical foundations is a more immediate and impactful direction for follow-up research.
>
> > The benchmarks are somewhat sparse, and some of them include results that have 100% test accuracy for many methods, so we cannot know which is better. Considering more difficult datasets would be more interesting.
>
> We thank the reviewer for the valuable suggestion. We are currently expanding our experiments to include more datasets, such as CIFAR and IMDb, in an updated version of the paper. These additions will provide a clearer comparison of our model's performance relative to the baselines and help highlight its strengths.
>
> > The maximum number of qubits considered is 10 if I am not mistaken, which is significantly smaller than what can be simulated with even small computational resources (20 is feasible in the noiseless setting).
>
> This is a good observation! A key strength of our approach is that it does not require scaling to 20 qubits to demonstrate strong performance. In fact, smaller models are sufficient to achieve meaningful results. The limitation to 12 qubits in our experiments is due once again to the current constraints of using PyTorch. In the future, more advanced software implementations will enable us to run models with larger qubit counts, potentially leading to even shallower circuits.

---

> > ### Comment · Reviewer_T8KM · 2024-11-26
> > **Reply to rebuttal**
> >
> > As other reviewers have stated, I still believe that your paper should demonstrate a clear performance benefit in a realistic setting (with noise and measurement errors and overhead taken into account) to be accepted to ICLR. I still believe it should not be too expensive to do, at least for the number of qubits you considered, therefore I keep my score.

---

### Official Review · Reviewer_bdvM · 2024-11-02

**Soundness:** 2
**Presentation:** 3
**Contribution:** 2
**Rating:** 3
**Confidence:** 3

**Summary:**

This paper presents the Hamiltonian classifier, a quantum machine learning approach that enhances the data encoding by mapping inputs to Pauli strings, and provides related proof-of-principle experiments to demonstrate the advantages.

**Strengths:**

it is well-written and clearly presents the concept and idea.

**Weaknesses:**

1. the similar idea was proposed in Post-variational quantum neural networks (https://arxiv.org/pdf/2307.10560)
2. missing some theoretical analysis of the performance of the proposed model.

**Questions:**

1. How to choose the {${P_j}$}$_{j=1}^p$ in Eq.12?
2. is there any theoretical analysis that the generalization error is bounded with respect to the number of finite terms $P$?
3. instead of using parametrized quantum circuit, whether is it enough to only optimize the the parameters in measurement (parametrized hamiltonian? like https://arxiv.org/pdf/2307.10560)?

---

> ### Author Response · Authors · 2024-11-20
>
> We thank the reviewer for their kind feedback and are pleased that they found the paper well-written and appreciated the clarity with which the concept and idea were presented.
>
> We now address the critiques raised by the reviewer in the following section.
>
> > the similar idea was proposed in Post-variational quantum neural networks (https://arxiv.org/pdf/2307.10560)
>
> We thank the reviewer for pointing out this related work. This work rightfully deserves to be cited in our discussion given its similarities, however, there are four major differences with our present work:
>
> **Input Encoding:** Unlike Huang & Rebentrost, whose method still relies on input-dependent controlled gates for encoding, our approach sidesteps the need for input encoding altogether. We represent data in a classical format, eliminating the need for additional quantum resources in this crucial step.
>
> **Post-Variational Circuits:** While Huang & Rebentrost propose the use of post-variational circuits as an alternative method to compute gradients, this concept is orthogonal to our work and can synergize with our proposal. We believe it holds potential as an extension for future research but does not take away from our core contributions.
>
> **Weighted Sum Technique:** The use of a weighted sum of Pauli measurements, similar to our formulation, is not novel in itself, as other works have also explored reweighting Pauli operators (https://arxiv.org/pdf/2306.00061). What sets our approach apart is the way we extract these coefficients directly from the data (SIM model). Additionally, we demonstrate that encoding data directly as a Hamiltonian provides sufficient information for classification without the need for pre-processing, as showcased by our HAM and PEFF models.
>
> **Experimental Setup:** Huang & Rebentrost focus on the Fashion-MNIST dataset, with pre-processing to reduce input size, and experiment with only a few hundred samples. In contrast, our work evaluates four distinct datasets, in each case working on the full dataset, employs very minimal pre-processing (tokenization or unrolling of images), and explores the method further with ablation studies. We believe this represents a more thorough experimental analysis.
>
> > missing some theoretical analysis of the performance of the proposed model.
>
> We acknowledge the reviewer’s concern regarding the absence of a detailed theoretical performance analysis. However, as this work focuses on a proof-of-concept demonstration, such theoretical analysis lies beyond its intended scope. To address this, we will enhance our experimental setup to provide a more comprehensive empirical evaluation. Additionally, we highlight this limitation and propose incorporating a rigorous theoretical analysis in future work.
>
> > How to choose the {P_i} in Eq.12?
>
> The selection of Pauli strings in our method is currently random, which has proven to be an effective heuristic. We recognize that the classical shadows method, as used by Huang and Rebentrost, could improve sample complexity and serves as a natural extension of our work. We will highlight this possibility in the paper’s future work section.
>
> Additionally, we conducted experiments with Pauli strings of limited locality that were not included in the paper. While these experiments were not a proper implementation of the classical shadows method, they did not show significant differences in downstream performance. This supports the practicality of our current random selection approach as a baseline.

---

### Official Review · Reviewer_Q4PZ · 2024-11-02

**Soundness:** 2
**Presentation:** 3
**Contribution:** 2
**Rating:** 3
**Confidence:** 4

**Summary:**

Summary: In this paper, the authors put forward a new quantum machine learning (QML) classifier. Their new approach leverages the variational quantum eigensolver algorithm to find a parametrized quantum circuit that classifies inputs based on their expectation value. Underlying their new approach is their idea to embed input states within a Hamiltonian.

**Strengths:**

Strengths: The paper is exceptionally well-written. The exposition is clear, the protocol is well explained, and the background well-done. The idea to embed states in a Hamiltonian and then leverage VQE is interesting as well.

**Weaknesses:**

Weaknesses: The article has three main weaknesses.
   - No theoretical performance guarantees: There are no theoretical results guaranteeing any a super-polynomial improvements over classical methods.
   - Weak results on simulated data: The new method is routinely outperformed by other methods including logistic regression on both text datasets. Sure, some of the competitors require more parameters, but none of the other models are honestly that big. These results would be far more compelling if there was a clear super-polynomial advantage with the new method.
   - No discussion of robustness to noise: This is a big one. There’s no discussion or evaluation of the methods robustness to noise. Unless you’re proposing a fault-tolerant algorithm, you need to discuss noise and compare your method’s performance in the presence of NISQ-era levels of noise to classical competitors.

**Questions:**

What happens to your method’s performance when it is run on noisy quantum hardware?

How resource-intensive is it to make your method fault-tolerant?

Why did you not try HAM and PEFF on the Fashion-MNIST dataset? Apologies if this was already covered, but I struggled to find it.

---

> ### Author Response · Authors · 2024-11-20
>
> We thank the reviewer for their thoughtful analysis and for recognizing the strengths of our work. We are delighted to hear that they found the paper well-written and the exposition clear, and we appreciate your acknowledgment of the thorough background and protocol explanation. We're also grateful for the positive remarks on the originality of our method.
> We now address the critiques raised and provide detailed responses to further clarify and improve our work.
>
> > No theoretical performance guarantees: There are no theoretical results guaranteeing any a super-polynomial improvements over classical methods.
>
> We acknowledge the reviewer's observation. It is important to note, however, that this is a broader challenge in the field of quantum computing, particularly for variational quantum algorithms (VQAs) such as our work. VQAs are widely recognized as a promising approach, particularly in quantum simulation, yet as of now, no conclusive theoretical quantum advantage has been established over classical methods. This lack of proven quantum advantage is not a barrier to publication, as shown by numerous peer-reviewed works (e.g., https://arxiv.org/abs/2006.14619, https://www.nature.com/articles/s41567-019-0648-8, https://aclanthology.org/2024.emnlp-main.1000.pdf). Moreover, as highlighted in Schuld & Killoran, 2022, the focus on “beating” classical machine learning should not overshadow contributions on practical implementations and novel paradigms. Our work aligns with this broader perspective.
>
> While this work does not focus on theoretical guarantees, we emphasize its practical contributions by providing an extensive empirical evaluation that demonstrates the efficacy of our method in diverse scenarios. Our approach offers significant advantages, including drastically reduced quantum encoding costs and shallower circuits, which are critical for near-term quantum devices. We will further strenghten the paper with additional experiments in the updated version.
>
> > Weak results on simulated data: The new method is routinely outperformed by other methods including logistic regression on both text datasets. Sure, some of the competitors require more parameters, but none of the other models are honestly that big. These results would be far more compelling if there was a clear super-polynomial advantage with the new method.
>
> The critique overlooks the key contribution of our work: enabling quantum algorithms to compete on even ground with classical methods on realistic, large-scale datasets. Competing "on even ground" with classical methods in this context is a significant achievement, given that most quantum machine learning models require dimensionality reduction or subsampling to run in a feasible amount of time.
>
> While our aim was to demonstrate basic functionality, the performance of our method can be enhanced in several ways, as we mention in the paper: employing nonlinear ansatze, performing (a hardware-efficient) input encoding in the variational part, or stacking multiple layers of parameterized measurements.
>
> > No discussion of robustness to noise: This is a big one. There’s no discussion or evaluation of the methods robustness to noise. Unless you’re proposing a fault-tolerant algorithm, you need to discuss noise and compare your method’s performance in the presence of NISQ-era levels of noise to classical competitors.
> How resource-intensive is it to make your method fault-tolerant?
>
> While robustness to noise is indeed critical for NISQ-era applications, it’s important to clarify that demonstrating NISQ-friendliness does not necessarily require running algorithms in a noisy simulation. This paper focuses on introducing a quantum machine learning method that is theoretically suited for NISQ devices, rather than presenting fault-tolerant algorithms. Evaluating noise resilience in detail is outside the scope of our current work, and we will certainly cover this in the future, along with theoretical guarantees. We will update the manuscript to explicitly note that our approach does not address fault tolerance or noise robustness.
>
> > Why did you not try HAM and PEFF on the Fashion-MNIST dataset?
>
> While the SIM model is lightweight, requiring a single set of measurements to be carried out to infer probability scores for all classes, HAM and PEFF would require simulating one Hamiltonian for each class, resulting in models with millions of parameters and long training times. We did not explore HAM and PEFF on the Fashion-MNIST dataset primarily due to the significant computational cost involved, as this would have limited our ability to perform an extensive evaluation, including hyperparameter tuning and training with multiple seeds. Instead, we focused on a narrower yet more thorough investigation, choosing to study in more detail the SIM model given its potential physical implementation.

---

> ### Comment · Reviewer_Q4PZ · 2024-11-25
>
> Thank you for taking the time to thoughtfully engage with my review. While I appreciate your response (and hope I understand it!), I disagree with a few key points. In particular, I still believe that the combination of: (i) not having a theoretical super-polynomial advantage over classical methods, (ii) not demonstrating compelling advantages over classical methods on toy error models, and (iii) not demonstrating any advantages to or near-peer performance with classical methods on experimental hardware precludes the paper from appearing at ICLR.
>
> The lack of (ii) and (iii) means that you have not placed the quantum classifier "on even ground" with classical methods. Without theoretical performance guarantees, even ground means either demonstrating near-peer performance of the classifier run on experimental hardware with mildly sophisticated classical methods, or near-peer performance in simulations with a fault-tolerant implementation of the classifier. Anything else is a proof of concept under ideal and completely unobtainable conditions for the quantum classifier.
>
> In light of my concerns, can you comment on any experimental results or even simulated results using a realistic noise model (i.e., incoherent + coherent errors with amplitude damping and measurement errors)?

---

### Note · Authors · 2024-12-16

**Comment:**

After receiving the feedback, we decided to withdraw and restructure the paper for future submissions. We thank all the reviewers and meta-reviewers involved in the process for their valuable feedback.

**Withdrawal Confirmation:**

I have read and agree with the venue's withdrawal policy on behalf of myself and my co-authors.